# Robust Bloom Filters for Large Multilabel Classification Tasks

**Moustapha Cissé**
LIP6, UPMC
Sorbonne Université
Paris, France
`first.last@lip6.fr`

**Nicolas Usunier**
UT Compiègne, CNRS
Heudiasyc UMR 7253
Compiègne, France
`nusunier@utc.fr`

**Thierry Artieres, Patrick Gallinari**
LIP6, UPMC
Sorbonne Université
Paris, France
`first.last@lip6.fr`

## Abstract

This paper presents an approach to multilabel classification (MLC) with a large number of labels. Our approach is a reduction to binary classification in which label sets are represented by low dimensional binary vectors. This representation follows the principle of Bloom filters, a space-efficient data structure originally designed for approximate membership testing. We show that a naive application of Bloom filters in MLC is not robust to individual binary classifiers' errors. We then present an approach that exploits a specific feature of real-world datasets when the number of labels is large: many labels (almost) never appear together. Our approach is provably robust, has sublinear training and inference complexity with respect to the number of labels, and compares favorably to state-of-the-art algorithms on two large scale multilabel datasets.

## 1 Introduction

Multilabel classification (MLC) is a classification task where each input may be associated to several class labels, and the goal is to predict the label set given the input. This label set may, for instance, correspond to the different topics covered by a text document, or to the different objects that appear in an image. The standard approach to MLC is the one-vs-all reduction, also called Binary Relevance (BR) [16], in which one binary classifier is trained for each label to predict whether the label should be predicted for that input. While BR remains the standard baseline for MLC problems, a lot of attention has recently been given to improve on it. The first main issue that has been addressed is to improve prediction performances at the expense of computational complexity by learning correlations between labels [5] [8], [9] or considering MLC as an unstructured classification problem over label sets in order to optimize the subset $0/1$ loss (a loss of 1 is incurred as soon as the method gets one label wrong) [16]. The second issue is to design methods that scale to a large number of labels (e.g. thousands or more), potentially at the expense of prediction performances, by learning compressed representations of labels sets with lossy compression schemes that are efficient when label sets have small cardinality [6]. We propose here a new approach to MLC in this latter line of work. A "MLC dataset" refers here to a dataset with a large number of labels (at least hundreds to thousands), in which the target label sets are smaller than the number of labels by one or several orders of magnitude, which is the common in large-scale MLC datasets collected from the Web.

The major difficulty in large-scale MLC problems is that the computational complexity of training and inference of standard methods is at least linear in the number of labels $L$. In order to scale better with $L$, our approach to MLC is to encode individual labels on $K$-sparse bit vectors of dimension $B$, where $B \ll L$, and use a disjunctive encoding of label sets (i.e. bitwise-OR of the codes of the labels that appear in the label set). Then, we learn one binary classifier for each of the $B$ bits of the coding vector, similarly to BR (where $K = 1$ and $B = L$). By setting $K > 1$, one can encode individual labels unambiguously on far less than $L$ bits while keeping the disjunctive encoding unambiguous

for a large number of labels sets of small cardinality. Compared to BR, our scheme learns only $B$ binary classifiers instead of $L$, while conserving the desirable property that the classifiers can be trained independently and thus in parallel, making our approach suitable for large-scale problems.

The critical point of our method is a simple scheme to select the $K$ representative bits (i.e. those set to 1) of each label with two desirable properties. First, the encoding of "relevant" label sets are unambiguous with the disjunctive encoding. Secondly, the decoding step, which recovers a label set from an encoding vector, is robust to prediction errors in the encoding vector: in particular, we prove that the number of incorrectly predicted labels is no more than twice the number of incorrectly predicted bits. Our (label) encoding scheme relies on the existence of *mutually exclusive* clusters of labels in real-life MLC datasets, where labels in different clusters (almost) never appear in the same label set, but labels from the same clusters can. Our encoding scheme makes that $B$ becomes smaller as more clusters of similar size can be found. In practice, a strict partitioning of the labels into mutually exclusive clusters does not exist, but it can be fairly well approximated by removing a few of the most frequent labels, which are then dealt with the standard BR approach, and clustering the remaining labels based on their co-occurrence matrix. That way, we can control the encoding dimension $B$ and deal with the computational cost/prediction accuracy tradeoff.

Our approach was inspired and motivated by Bloom filters [2], a well-known space-efficient randomized data structure designed for approximate membership testing. Bloom filters use exactly the principle of encoding objects (in our case, labels) by $K$-sparse vectors and encode a set with the disjunctive encoding of its members. The filter can be queried with one object and the answer is correct up to a small error probability. The data structure is randomized because the representative bits of each object are obtained by random hash functions; under uniform probability assumptions for the encoded set and the queries, the encoding size $B$ of the Bloom filter is close to the information theoretic limit for the desired error rate. Such "random" Bloom filter encodings are our main baseline, and we consider our approach as a new design of the hash functions and of the decoding algorithm to make Bloom filter robust to errors in the encoding vector. Some background on (random) Bloom filters, as well as how to apply them for MLC is given in the next section. The design of hash functions and the decoding algorithm are then described in Section 3, where we also discuss the properties of our method compared to related works of [12, 15, 4]. Finally, in Section 4, we present experimental results on two benchmark MLC datasets with a large number of classes, which show that our approach obtains promising performances compared to existing approaches.

## 2  Bloom Filters for Multilabel Classification

Our approach is a reduction from MLC to binary classification, where the rules of the reduction follow a scheme inspired by the encoding/decoding of sets used in Bloom filters. We first describe the formal framework to fix the notation and the goal of our approach, and then give some background on Bloom filters. The main contribution of the paper is described in the next section.

**Framework**  Given a set of labels $\mathcal{L}$ of size $L$, MLC is the problem of learning a prediction function $c$ that, for each possible input $x$, predicts a subset of $\mathcal{L}$. Throughout the paper, the letter $y$ is used for label sets, while the letter $\ell$ is used for individual labels. Learning is carried out on a training set $((x_1, y_1), ..., (x_n, y_n))$ of inputs for which the desired label sets are known; we assume the examples are drawn i.i.d. from the data distribution $\mathcal{D}$.

A reduction from MLC to binary classification relies on an encoding function $\mathbf{e} : y \subseteq \mathcal{L} \mapsto (e_1(y), ..., e_B(y)) \in \{0, 1\}^B$, which maps subsets of $\mathcal{L}$ to bit vectors of size $B$. Then, each of the $B$ bits are learnt independently by training a sequence of binary classifiers $\hat{\mathbf{e}} = (\hat{e}_1, ..., \hat{e}_B)$, where each $\hat{e}_j$ is trained on $((x_1, e_j(y_1)), ..., (x_n, e_j(y_n)))$. Given a new instance $x$, the encoding $\hat{\mathbf{e}}(x)$ is predicted, and the final multilabel classifier $c$ is obtained by decoding $\hat{\mathbf{e}}(x)$, i.e. $\forall x, c(x) = d(\hat{\mathbf{e}}(x))$. The goal of this paper is to design the encoding and decoding functions so that two conditions are met. First, the code size $B$ should be small compared to $L$, in order to improve the computational cost of training and inference relatively to BR. Second, the reduction should be robust in the sense that the final performance, measured by the expected Hamming loss $\mathbb{H}^L(c)$ between the target label sets $y$ and the predictions $c(x)$ is not much larger than $\mathbb{H}^B(\hat{\mathbf{e}})$, the average error of the classifiers we learn. Using $\Delta$ to denote the symmetric difference between sets, $\mathbb{H}^L$ and $\mathbb{H}^B$ are defined by:

$$\mathbb{H}^L(c) = \mathbb{E}_{(x,y)\sim\mathcal{D}}\left[\frac{|c(x)\Delta y|}{L}\right] \quad \text{and} \quad \mathbb{H}^B(\hat{\mathbf{e}}) = \frac{1}{B}\sum_{j=1}^{B}\mathbb{E}_{(x,y)\sim\mathcal{D}}\left[\mathbf{1}_{\{e_j(y)\neq\hat{e}_j(y)\}}\right] . \quad (1)$$

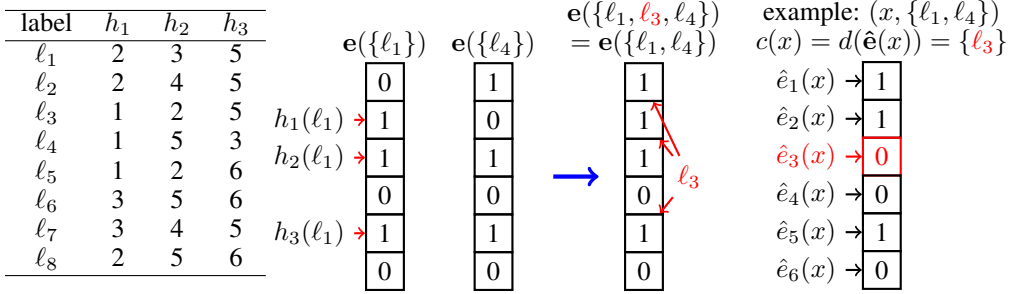

Figure 1: Examples of a Bloom filter for a set $\mathcal{L} = \{\ell_1, ..., \ell_8\}$ with 8 elements, using 3 hash functions and 6 bits). *(left)* The table gives the hash values for each label. *(middle-left)* For each label, the hash functions give the index of the bits that are set to 1 in the 6-bit boolean vector. The examples of the encodings for $\{\ell_1\}$ and $\{\ell_4\}$ are given. *(middle-right)* Example of a false positive: the representation of the subset $\{\ell_1, \ell_4\}$ includes all the representative bits of label $\ell_3$ so that is $\ell_3$ would be decoded erroneously. *(right)* Example of propagation of errors: a single erroneous bit in the label set encoding, together with a false positive, leads to three label errors in the final prediction.

**Bloom Filters** Given the set of labels $\mathcal{L}$, a Bloom filter (BF) of size $B$ uses $K$ hash functions from $\mathcal{L}$ to $\{1, ..., B\}$, which we denote $h_k : \mathcal{L} \to \{1, ..., B\}$ for $k \in \{1, ..., K\}$ (in a standard approach, each value $h_k(\ell)$ is chosen uniformly at random in $\{1, ..., B\}$). These hash functions define the representative bits (i.e. non-zero bits) of each label: each singleton $\{\ell\}$ for $\ell \in \mathcal{L}$ is encoded by a bit vector of size $B$ with at most $K$ non-zero bits, and each hash function gives the index of one of these nonzero bits in the bit vector. Then, the Bloom filter encodes a subset $y \subseteq \mathcal{L}$ by a bit vector of size $B$, defined by the bitwise OR of the bit vectors of the elements of $y$. Given the encoding of a set, the Bloom filter can be queried to test the membership of any label $\ell$; the filter answers positively if all the representative bits of $\ell$ are set to 1, and negatively otherwise. A negative answer of the Bloom filter is always correct; however, the bitwise OR of label set encodings leads to the possibility of *false positives*, because even though any two labels have different encodings, the representative bits of one label can be included in the union of the representative bits of two or more other labels. Figure 1 (left) to (middle-right) give representative examples of the encoding/querying scheme of Bloom filters and an example of false positive.

**Bloom Filters for MLC** The encoding and decoding schemes of BFs are appealing to define the encoder $\mathbf{e}$ and the decoder $d$ in a reduction of MLC to binary classification (decoding consists in querying each label), because they are extremely simple and computationally efficient, but also because, if we assume that $B \ll L$ and that the random hash functions are perfect, then, given a random subset of size $C \ll L$, the false positive rate of a BF encoding this set is in $O\left(\left(\frac{1}{2}\right)^{\frac{C}{B}\ln(2)}\right)$ for the optimal number of hash functions. This rate is, up to a constant factor, the information theoretic limit [3]. Indeed, as shown in Section 4 the use of Bloom filters with random hash functions for MLC (denoted S-BF for Standard BF hereafter) leads to rather good results in practice.

Nonetheless, there is much room for improvement with respect to the standard approach above. First, the distribution of label sets in usual MLC datasets is far from uniform. On the one hand, this leads to a substantial increase in the error rate of the BF compared to the theoretical calculation, but, on the other hand, it is an opportunity to make sure that false positive answers only occur in cases that are detectable from the observed distribution of label sets: if $y$ is a label set and $\ell \notin y$ is a false positive given $\mathbf{e}(y)$, $\ell$ can be detected as a false positive if we know that $\ell$ never (or rarely) appears together with the labels in $y$. Second and more importantly, the decoding approach of BFs is far from robust to errors in the predicted representation. Indeed, BFs are able to encode subsets on $B \ll L$ bits because each bit is representative for several labels. In the context of MLC, the consequence is that any single bit incorrectly predicted may include in (or exclude from) the predicted label set all the labels for which it is representative. Figure 1 (right) gives an example of the situation, where a single error in the predicted encoding, added with a false positive, results in 3 errors in the final prediction. Our main contribution, which we detail in the next section, is to use the non-uniform distribution of label sets to design the hash functions and a decoding algorithm to make sure that any incorrectly predicted bit has a limited impact on the predicted label set.

# 3 From Label Clustering to Hash Functions and Robust Decoding

We present a new method that we call Robust Bloom Filters (R-BF). It improves over random hash functions by relying on a structural feature of the label sets in MLC datasets: many labels are never observed in the same target set, or co-occur with a probability that is small enough to be neglected. We first formalize the structural feature we use, which is a notion of mutually exclusive clusters of labels, then we describe the hash functions and the robust decoding algorithm that we propose.

## 3.1 Label Clustering

The strict formal property on which our approach is based is the following: given $P$ subsets $\mathcal{L}_1, ..., \mathcal{L}_P$ of $\mathcal{L}$, we say that $(\mathcal{L}_1, ..., \mathcal{L}_P)$ are *mutually exclusive clusters* if no target set contains labels from more than one of each $\mathcal{L}_p, p = 1..P$, or, equivalently, if the following condition holds:

$$\forall p \in \{1, ..., P\}, \mathbb{P}_{y \sim \mathcal{D}_{\mathcal{Y}}} \Big( \ \big(y \cap \mathcal{L}_p \neq \emptyset\big) \ \text{ and } \ \big(y \cap \bigcup_{p' \neq p} \mathcal{L}_{p'} \neq \emptyset\big) \ \Big) = 0 \,. \tag{2}$$

where $\mathcal{D}_{\mathcal{Y}}$ is the marginal distribution over label sets. For the disjunctive encoding of Bloom filters, this assumption implies that if we design the hash functions such that the false positives for a label set $y$ belong to a cluster that is mutually exclusive with (at least one) label in $y$, then the decoding step can detect and correct it. To that end, it is sufficient to ensure that for each bit of the Bloom filter, all the labels for which this bit is representative belong to mutually exclusive clusters. This will lead us to a simple two-step decoding algorithm cluster identification/label set prediction in the cluster. In terms of compression ratio $\frac{B}{L}$, we can directly see that the more mutually exclusive clusters, the more labels can share a single bit of the Bloom filter. Thus, more (balanced) mutually exclusive clusters will result in smaller encoding vectors $B$, making our method more efficient overall.

This notion of mutually exclusive clusters is much stronger than our basic observation that some pair of labels rarely or never co-occur with each other, and in practice it may be difficult to find a partition of $\mathcal{L}$ into mutually exclusive clusters because the co-occurrence graph of labels is connected. However, as we shall see in the experiments, after removing the few most central labels (which we call *hubs*, and in practice roughly correspond to the most frequent labels), the labels can be clustered into (almost) mutually exclusive labels using a standard clustering algorithm for weighted graph.

In our approach, the hubs are dealt with outside the Bloom filter, with a standard binary relevance scheme. The prediction for the remaining labels is then constrained to predict labels from at most one of the clusters. From the point of view of prediction performance, we loose the possibility of predicting arbitrary label sets, but gain the possibility of correcting a non-negligible part of the incorrectly predicted bits. As we shall see in the experiments, the trade-off is very favorable. We would like to note at this point that dealing with the *hubs* or the most frequent labels with binary relevance may not particularly be a drawback of our approach: the occurrence probabilities of the labels is long-tailed, and the first few labels may be sufficiently important to deserve a special treatment. What really needs to be compressed is the large set of labels that occur rarely.

To find the label clustering, we first build the co-occurrence graph and remove the hubs using the degree centrality measure. The remaining labels are then clustered using Louvain algorithm [1]; to control the number of clusters, a maximum size is fixed and larger clusters are recursively clustered until they reach the desired size. Finally, to obtain (almost) balanced clusters, the smallest clusters are merged. Both the number of hubs and the cluster size are parameters of the algorithm, and, in Section 4, we show how to choose them before training at negligible computational cost.

## 3.2 Hash functions and decoding

From now on, we assume that we have access to a partition of $\mathcal{L}$ into mutually exclusive clusters (in practice, this corresponds to the labels that remain after removal of the hubs).

**Hash functions** Given the parameter $K$, constructing $K$-sparse encodings follows two conditions:

1. two labels from the same cluster cannot share any representative bit;
2. two labels from different clusters can share at most $K - 1$ representative bits.

| bit index | representative for labels | bit index | representative for labels | cluster index | labels in cluster | cluster index | labels in cluster |
|---|---|---|---|---|---|---|---|
| 1 | $\{1,2,3,4,5\}$ | 7 | $\{16,17,18,19,20\}$ | 1 | $\{1,15\}$ | 9 | $\{9,23\}$ |
| 2 | $\{1,6,7,8,9\}$ | 8 | $\{16,21,22,23,24\}$ | 2 | $\{2,16\}$ | 10 | $\{10,24\}$ |
| 3 | $\{2,6,10,11,12\}$ | 9 | $\{17,21,25,26,27\}$ | 3 | $\{3,17\}$ | 11 | $\{11,25\}$ |
| 4 | $\{3,7,10,13,14\}$ | 10 | $\{18,22,25,28,29\}$ | 4 | $\{4,18\}$ | 12 | $\{12,26\}$ |
| 5 | $\{4,8,11,13,15\}$ | 11 | $\{19,23,26,28,30\}$ | 5 | $\{5,19\}$ | 13 | $\{13,27\}$ |
| 6 | $\{5,9,12,14,15\}$ | 12 | $\{20,24,27,29,30\}$ | 6 | $\{6,20\}$ | 14 | $\{14,28\}$ |
| | | | | 7 | $\{7,21\}$ | 15 | $\{15,29\}$ |
| | | | | 8 | $\{8,22\}$ | | |

Figure 2: Representative bits for 30 labels partitioned into $P = 15$ mutually exclusive label clusters of size $R = 2$, using $K = 2$ representative bits per label and batches of $Q = 6$ bits. The table on the right gives the label clustering. The injective mapping between labels and subsets of bits is defined by $\mathbf{g} : \ell \mapsto \{g_1(\ell) = (1+\ell)/6, g_2(\ell) = 1+\ell \mod 6\}$ for $\ell \in \{1, ..., 15\}$ and, for $\ell \in \{15, ..., 30\}$, it is defined by $\ell \mapsto \{(6 + g_1(\ell - 15), 6 + g_1(\ell - 15)\}$.

Finding an encoding that satisfies the conditions above is not difficult if we consider, for each label, the set of its representative bits. In the rest of the paragraph, we say that a bit of the Bloom filter "is used for the encoding of a label" when this bit may be a representative bit of the label. If the bit "is not used for the encoding of a label", then it cannot be a representative bit of the label.

Let us consider the $P$ mutually exclusive label clusters, and denote by $R$ the size of the largest cluster. To satisfy Condition 1., we find an encoding on $B = R.Q$ bits for $Q \geq K$ and $P \leq \binom{Q}{K}$ as follows. For a given $r \in \{1, ..., R\}$, the $r$-th batch of $Q$ successive bits (i.e. the bits of index $(r-1)Q + 1, (r-1)Q + 2, ..., rQ$) is used only for the encoding of the $r$-th label of each cluster. That way, each batch of $Q$ bits is used for the encoding of a single label per cluster (enforcing the first condition) but can be used for the encoding of $P$ labels overall. For the Condition 2., we notice that given a batch of $Q$ bits, there are $\binom{Q}{K}$ different subsets of $K$ bits. We then injectively map the (at most) $P$ labels to the subsets of size $K$ to define the $K$ representative bits of these labels. In the end, with a Bloom filter of size $B = R.Q$, we have $K$-sparse encodings that satisfy the two conditions above for $L \leq R.\binom{Q}{K}$ labels partitioned into $P \leq \binom{Q}{K}$ mutually exclusive clusters of size at most $R$. Figure 2 gives an example of such an encoding. In the end, the scheme is most efficient (in terms of the compression ratio $B/L$) when the clusters are perfectly balanced and when $P$ is exactly equal to $\binom{Q}{K}$ for some $Q$. For instance, for $K = 2$ that we use in our experiments, if $P = \frac{Q(Q+1)}{2}$ for some integer $Q$, and if the clusters are almost perfectly balanced, then $B/L \approx \sqrt{2/P}$. The ratio becomes more and more favorable as both $Q$ increases and $K$ increases up to $Q/2$, but the number of different clusters $P$ must also be large. Thus, the method should be most efficient on datasets with a very large number of labels, assuming that $P$ increases with $L$ in practice.

**Decoding and Robustness** We now present the decoding algorithm, followed by a theoretical guarantee that each incorrectly predicted bit in the Bloom filter cannot imply more than 2 incorrectly predicted labels.

Given an example $x$ and its predicted encoding $\hat{\mathbf{e}}(x)$, the predicted label set $d(\hat{\mathbf{e}}(x))$ is computed with the following two-step process, in which we say that a bit is "representative of one cluster" if it is a representative bit of one label in the cluster:

a. *(Cluster Identification)* For each cluster $\mathcal{L}_p$, compute its cluster score $s_p$ defined as the number of its representative bits that are set to 1 in $\hat{\mathbf{e}}(x)$. Choose $\mathcal{L}_{\hat{p}}$ for $\hat{p} \in \arg\max_{p \in \{1,...,P\}} s_p$;

b. *(Label Set Prediction)* For each label $\ell \in \mathcal{L}_{\hat{p}}$, let $s'_\ell$ be the number of representative bits of $\ell$ set to 1 in $\hat{\mathbf{e}}(x)$; add $\ell$ to $d(\hat{\mathbf{e}}(x))$ with probability $\frac{s'_\ell}{K}$.

In case of ties in the cluster identification, the tie-breaking rule can be arbitrary. For instance, in our experiments, we use logistic regression as base learners for binary classifiers, so we have access to posterior probabilities of being 1 for each bit of the Bloom filter. In case of ties in the cluster identification, we restrict our attention to the clusters that maximize the cluster score, and we recompute their cluster scores using the posterior probabilities instead of the binary decision. The

cluster which maximizes the new cluster score is chosen. The choice of a randomized prediction for the labels avoids a single incorrectly predicted bit to result in too many incorrectly predicted labels. The robustness of the encoding/decoding scheme is proved below:

**Theorem 1** *Let the label set $\mathcal{L}$ , and let $(\mathcal{L}_1, ..., \mathcal{L}_P)$ be a partition of $\mathcal{L}$ satisfying (2). Assume that the encoding function satisfies Conditions 1. and 2., and that decoding is performed in the two-step process a.-b. Then, using the definitions of $\mathbb{H}^L$ and $\mathbb{H}^B$ of (1), we have:*

$$\mathbb{H}^L(d \circ \hat{\mathbf{e}}) \leq \frac{2B}{L} \mathbb{H}^B(\hat{\mathbf{e}})$$

*for a $K$-sparse encoding, where the expectation in $\mathbb{H}^L$ is also taken over the randomized predictions.*

*Sketch of proof* Let $(x, y)$ be an example. We compare the expected number of incorrectly predicted labels $H^L(y, d(\hat{\mathbf{e}}(x))) = \mathbb{E}\big[|d(\hat{\mathbf{e}}(x)) \, \Delta \, y|\big]$ (expectation taken over the randomized prediction) and the number of incorrectly predicted bits $H^B(\hat{\mathbf{e}}(x), \mathbf{e}(y)) = \sum_{j=1}^{B} \mathbf{1}_{\{\hat{e}_j(x) \neq e_j(y)\}}$. Let us denote by $p^*$ the index of the cluster in which $y$ is included, and $\hat{p}$ the index of the cluster chosen in step a. We consider the two following cases:

$\hat{p} = p^*$: if the cluster is correctly identified then each incorrectly predicted bit that is representative for the cluster costs $\frac{1}{K}$ in $H^L(y, d(\hat{\mathbf{e}}(x)))$. All other bits do not matter. We thus have $H^L(y, d(\hat{\mathbf{e}}(x))) \leq \frac{1}{K} H^B(\hat{\mathbf{e}}(x), \mathbf{e}(y))$.

$\hat{p} \neq p^*$: If the cluster is not correctly identified, then $H^L(y, d(\hat{\mathbf{e}}(x)))$ is the sum of (1) the number of labels that should be predicted but are not ($|y|$), and (2) the labels that are in the predicted label set but that should not. To bound the ratio $\frac{H^L(y, d(\hat{\mathbf{e}}(x)))}{H^B(\hat{\mathbf{e}}(x), \mathbf{e}(y))}$, we first notice that there are at least as much representative bits predicted as 1 for $\mathcal{L}_{\hat{p}}$ than for $\mathcal{L}_{p^*}$. Since each label of $\mathcal{L}_{\hat{p}}$ shares at most $K - 1$ representative bits with a label of $\mathcal{L}_{p^*}$, there are at least $|y|$ incorrect bits. Moreover, the maximum contribution to labels predicted in the incorrect cluster by correctly predicted bits is at most $\frac{K-1}{K}|y|$. Each additional contribution of $\frac{1}{K}$ in $H^L(y, d(\hat{\mathbf{e}}(x)))$ comes from a bit that is incorrectly predicted to 1 instead of 0 (and is representative for $\mathcal{L}_{\hat{p}}$). Let us denote by $k$ the number of such contributions. Then, the most defavorable ratio $\frac{H^L(y, d(\hat{\mathbf{e}}(x)))}{H^B(\hat{\mathbf{e}}(x), \mathbf{e}(y))}$ is smaller than $\max\limits_{k \geq 0} \frac{\frac{k}{K} + |y|(1 + \frac{K-1}{K})}{\max(|y|, k)} = \frac{\frac{|y|}{K} + |y|(1 + \frac{K-1}{K})}{|y|} = 2$.

Taking the expectation over $(x, y)$ completes the proof ($\frac{B}{L}$ comes from normalization factors). □

### 3.3 Comparison to Related Works

The use of correlations between labels has a long history in MLC [11] [8] [14], but correlations are most often used to improve prediction performances at the expense of computational complexity through increasingly complex models, rather than to improve computational complexity using strong negative correlations as we do here.

The most closely related works to ours is that of Hsu et al. [12], where the authors propose an approach based on compressed sensing to obtain low-dimension encodings of label sets. Their approach has the advantage of a theoretical guarantee in terms of regret (rather than error as we do), without strong structural assumptions on the label sets; the complexity of learning scales in $O(C \ln(L))$ where $C$ is the number of labels in label sets. For our approach, since $\binom{Q}{\frac{Q}{2}} \underset{Q \to \infty}{\sim} 4^{Q/2}/\sqrt{8\pi Q}$, it could be possible to obtain a logarithmic rate under the rather strong assumption that the number of clusters $P$ increases linearly with $L$. As we shall see in our experiments, however, even with a rather large number of labels (e.g. 1 000), the asymptotic logarithmic rate is far from being achieved for all methods. In practice, the main drawback of their method is that they need to know the size of the label set to predict. This is an extremely strong requirement when classification decisions are needed (less strong when only a ranking of the labels is needed), in contrast to our method which is inherently designed for classification.

Another related work is that of [4], which is based on SVD for dimensionality reduction rather than compressed sensing. Their method can exploit correlations between labels, and take classification decisions. However, their approach is purely heuristic, and no theoretical guarantee is given.

Figure 3: *(left)* Unrecoverable Hamming loss (UHL) due to label clustering of the R-BF as a function of the code size $B$ on *RCV-Industries* (similar behavior on the *Wikipedia1k* dataset). The optimal curve represents the best UHL over different settings (number of hubs,max cluster size) for a given code size. *(right)* Hamming loss vs code size on *RCV-Industries* for different methods.

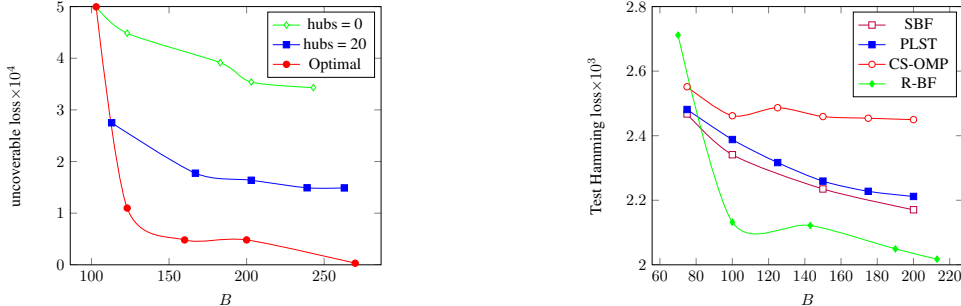

## 4  Experiments

We performed experiments on two large-scale real world datasets: *RCV-Industries*, which is a subset of the RCV1 dataset [13] that considers the industry categories only (we used the first testing set file from the RCV1 site instead of the original training set since it is larger), and *Wikipedia1k*, which is a subsample of the wikipedia dataset release of the 2012 large scale hierarchical text classification challenge [17]. On both datasets, the labels are originally organized in a hierarchy, but we transformed them into plain MLC datasets by keeping only leaf labels. For *RCV-Industries*, we obtain 303 labels for $72,334$ examples. The average cardinality of label sets is $1.73$ with a maximum of 30; 20% of the examples have label sets of cardinality $\geq 2$. For *Wikipedia1k*, we kept the $1,000$ most represented leaf labels, which leads to $110,530$ examples with an average label set cardinality of $1.11$ (max. 5). 10% of the examples have label sets of cardinality $\geq 2$.

We compared our methods, the standard (i.e. with random hash function) BF (S-BF) and the Robust BF (R-BF) presented in section 3, to binary relevance (BR) and to three MLC algorithms designed for MLC problems with a large number of labels: a pruned version of BR proposed in [7] (called BR-*Dekel* from now on), the compressed sensing approach (CS) of [12] and the principal label space transformation (PLST) [4]. BR-*Dekel* consists in removing from the prediction all the labels whose probability of a true positive (PTP) on the validation set is smaller than the probability of a false positive (PFP). To control the code size $B$ in BR-*Dekel*, we rank the labels based on the ratio $PTP/PFP$ and keep the top $B$ labels. In that case, the inference complexity is similar to BF models, but the training complexity is still linear in $L$. For CS, following [4], we used orthogonal matching poursuit (CS-OMP) for decoding and selected the number of labels to predict in the range $\{1, 2, \ldots, 30\}$, on the validation set. For S-BF, the number of (random) hash functions $K$ is also chosen on the validation set among $\{1, 2, \ldots, 10\}$. For R-BF, we use $K = 2$ hash functions.

The code size $B$ can be freely set for all methods except for Robust BF, where different settings of the maximum cluster size and the number of hubs may lead to the same code size. Since the use of a label clustering in R-BF leads to unrecoverable errors even if the classifiers perform perfectly well (because labels of different clusters cannot be predicted together), we chose the max cluster size among $\{10, 20, \ldots, 50\}$ and the number of hubs (among $\{0, 10, 20, 30, \ldots, 100\}$ for *RCV-Industries* and $\{0, 50, 100, \ldots, 300\}$ for *Wikipedia1k*) that minimize the resulting unrecoverable Hamming loss (UHL), computed on the train set. Figure 3 *(left)* shows how the UHL naturally decreases when the number of hubs increases since then the method becomes closer to BR, but at the same time the overall code size $B$ increases because it is the sum of the filter's size and the number of hubs. Nonetheless, we can observe on the figure that the UHL rapidly reaches a very low value, confirming that the label clustering assumption is reasonable in practice.

All the methods involve training binary classifiers or regression functions. On both datasets, we used linear functions with $L_2$ regularization (the global regularization factor in PLST and CS-OMP, as well as the regularization factor of each binary classifier in BF and BR approaches, were chosen on the validation set among $\{0, 0.1, \ldots, 10^{-5}\}$), and unit-nom normalized TF-IDF features. We used the Liblinear [10] implementation of logistic regression as base binary classifier.

Table 1: Test Hamming loss (HL, in %), micro (m-F1) and macro (M-F1) F1-scores. $B$ is code size. The results of the significance test for a $p$-value less than $5\%$ are denoted † to indicate the best performing method using the same $B$ and ∗ to indicate the best performing method overall.

| Classifier | $B$ | HL | m-F1 | M-F1 | $B$ | HL | m-F1 | M-F1 |
|---|---|---|---|---|---|---|---|---|
| | | RCV-Industries | | | | Wikipedia1K | | |
| BR | 303 | 0.200* | 72.43* | 47.82* | 1000 | 0.0711 | 55.96 | 34.7 |
| BR-Dekel | 150 | 0.308 | 46.98 | 30.14 | 250 | 0.0984 | 22.18 | 12.16 |
| | 200 | 0.233 | 65.78 | 40.09 | 500 | 0.0868 | 38.33 | 24.52 |
| S-BF | 150 | 0.223 | 67.45 | 40.29 | 250 | 0.0742 | 53.02 | 31.41 |
| | 200 | 0.217 | 68.32 | 40.95 | 500 | 0.0734 | 53.90 | 32.57 |
| R-BF | 150 | $0.210^{\dagger}$ | $71.31^{\dagger}$ | 43.44 | 240 | $0.0728^{\dagger}$ | 55.85 | 34.65 |
| | 200 | $0.205^{\dagger}$ | $71.86^{\dagger}$ | 44.57 | 500 | $0.0705^{\dagger *}$ | 57.31 | 36.85 |
| CS-OMP | 150 | 0.246 | 67.59 | $45.22^{\dagger}$ | 250 | 0.0886 | $57.96^{\dagger}$ | $41.84^{\dagger}$ |
| | 200 | 0.245 | 67.71 | $45.82^{\dagger}$ | 500 | 0.0875 | $58.46^{\dagger *}$ | $42.52^{\dagger *}$ |
| PLST | 150 | 0.226 | 68.87 | 32.36 | 250 | 0.0854 | 42.45 | 09.53 |
| | 200 | 0.221 | 70.35 | 40.78 | 500 | 0.0828 | 45.95 | 16.73 |

**Results** Table 1 gives the test performances of all the methods on both datasets for different code sizes. We are mostly interested in the Hamming loss but we also provide the micro and macro F-measure. The results are averaged over 10 random splits of train/validation/test of the datasets, respectively containing $50\%/25\%/25\%$ of the data. The standard deviations of the values are negligible (smaller than $10^{-3}$ times the value of the performance measure). Our BF methods seem to clearly outperform all other methods and R-BF yields significant improvements over S-BF. On *Wikipedia1k*, with 500 classifiers, the Hamming loss (in %) of S-BF is 0.0734 while it is only 0.0705 for RBF. This performance is similar to that of BR's (0.0711) which uses twice as many classifiers. The simple pruning strategy *BR-Dekel* is the worst baseline on both datasets, confirming that considering all classes is necessary on these datasets. CS-OMP reaches a much higher Hamming loss (about $23\%$ worst than BR on both datasets when using $50\%$ less classifiers). CS-OMP achieves the best performance on the macro-F measure though. This is because the size of the predicted label sets is fixed for CS, which increases recall but leads to poor precision. We used OMP as decoding procedure for CS since it seemed to perform better than Lasso and Correlation decoding (CD)[12]( for instance, on *RCV-Industries* with a code size of 500, OMP achieves a Hamming loss of 0.0875 while the Hamming loss is 0.0894 for Lasso and 0.1005 for CD). PLST improves over CS-OMP but its performances are lower than those of S-BF (about $3.5\%$ on RCV-industries and $13\%$ and Wikipedia when using $50\%$ less classifiers than BR). The macro F-measure indicates that PLST likely suffers from class imbalance (only the most frequent labels are predicted), probably because the label set matrix on which SVD is performed is dominated by the most frequent labels. Figure 3 (right) gives the general picture of the Hamming loss of the methods on a larger range of code sizes. Overall, R-BF has the best performances except for very small code sizes because the UHL becomes too high.

**Runtime analysis** Experiments were performed on a computer with 24 intel Xeon 2.6 GHz CPUs. For all methods, the overall training time is dominated by the time to train the binary classifiers or regressors, which depends linearly on the code size. For test, the time is also dominated by the classifiers' predictions, and the decoding algorithm of R-BF is the fastest. For instance, on *Wikipedia1k*, training one binary classifier takes $12.35s$ on average, and inference with one classifier (for the whole test dataset) takes $3.18s$. Thus, BR requires about 206 minutes ($1000 \times 12.35s$) for training and $53m$ for testing on the whole test set. With $B = 500$, R-BF requires about half that time, including the selection of the number of hubs and the max. cluster size at training time, which is small (computing the UHL of a R-BF configuration takes $9.85s$, including the label clustering step, and we try less than 50 of them). For the same $B$, encoding for CS takes $6.24s$ and the SVD in PSLT takes $81.03s$, while decoding takes $24.39s$ at test time for CS and $7.86s$ for PSLT.

**Acknowledgments**

This work was partially supported by the French ANR as part of the project Class-Y (ANR-10-BLAN-02) and carried out in the framework of the Labex MS2T (ANR-11-IDEX-0004-02).

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
