[Reviews · NeurIPS 2013]

Submitted by Assigned_Reviewer_5

I think this is an interesting paper. The connection to bloom filters took a while for me to parse through, and the abstract was far from clear. However, I think the presentation is adequate for a conference paper. I like the emphasis on sub-linear classification cost, but wish this has been explored more explicitly in the experimental results. Overall, the experimental results are strong, and are supported by useful theory.
Summary: The paper proposes a novel mechanism for multi-label classification, inspired by bloom filters (I thought of it as basically a theoretically grounded way to use random projections for label clustering). The paper then goes on to show how this framework can be augmented using label clustering methods to be more robust when label distributions are non-uniform (as is normally the case). Empirical results are strong.

Submitted by Assigned_Reviewer_6

This is a well-written paper that makes an interesting contribution to an important topic. In fact, the idea of using Bloom filters for dimensionality reduction in multi-label classification is intriguing. This seems to be a companion paper of another paper, in which this idea has already been introduced. The contribution of this paper is to improve the encoding scheme used in the Bloom filter. To this end, the structure of the label space is exploited by means of clustering methods. As claimed by the authors, this encoding makes the filter more robust.

While I'm essentially fine with the idea of the method and theoretical part of the paper, I am much less convinced of the empirical study. First, given the goal of efficiency, one would certainly expect an empirical analysis of runtime in the experimental part, not only predictive accuracy.

Moreover, the two data sets used in the experiments are somewhat debatable, especially due to their extremely low label cardinality (it would be interesting to see the marginal label distribution). In addition to the sparsity, the distribution will probably be skewed, so that most of the labels will almost never occur. In fact, one may wonder whether such data sets are sufficiently representative to provide an idea of how the authors' approach performs. Indeed, the performance of this approach (like the performance of other compression schemes) strongly depends on the label distribution, which is very specific for the data sets used.

Moreover, losses like Hamming, which the authors focus on, are debatable in such cases, especially since always predicting negative will already yield an extremely low loss. For example, the author report a Hamming loss of 0.000734 for SBF and "only" 0.000705 for RBF. Is this tiny difference of any meaning? Hard to say, especially since no standard deviations are reported.
Summary: Nice paper that definitely holds promise. While the theoretical part is fine, the empirical one is less convincing.

Submitted by Assigned_Reviewer_7

The paper proposes a multi-label classification algorithm: a large number of classes is hashed down into a smaller number of classifiers, via Bloom-filter-like rule. The hashing function is chosen to be consonant with co-occurance of rare class labels, for better decoding.

Significance: The idea of mapping multi-label classification into Bloom filters is insightful, and deserves to be published. It's interesting that a Naive Bloom filter does not work well -- the experimental results show that you need to carefully chose the Bloom filters (rather generating them randomly). The experimental results show that you can save ~2x in number of output classes --- this is a good result, although not earth-shattering.

Novelty: this idea is new, as far as I know. The authors correctly cite to paper reference [12] as the closest idea.

Clarify: The paper is quite clear.

Quality: I am concerned about the comparison to paper reference [12], which proposes learning a compressed version of the output labels, and then use various compressed sensing decoding techniques to recover the sparse output labels. The current paper uses Orthogonal Matching Pursuit to decode the compressed labels. But, in paper [12], OMP sometimes is substantially worse than using CoSaMP or LASSO. In the feedback, the authors say they will squeeze in some more results.
Summary: A nice idea to use Bloom filters to perform multi-label classification.
Author Feedback

Author rebuttal: We first thank the reviewers for their helpful reviews. We will work on the abstract to clarify it. We now answer the reviewers' main points, and will modify the paper accordingly if it is accepted.

Assigned_reviewer6:

- runtime analysis: we agree with the reviewer that a quantitative analysis should be added. For all methods, the runtime of building the models (resp. performing prediction) is dominated by the training (resp. prediction) time of the binary classifiers/regressors. The encoding step, the label clustering as well as finding the optimal configuration for the Robust Bloom filter (RBF) are negligible. Thus, the runtime for RBF is roughly linear in the code size (while the runtime of binary relevance (BR) is linear in the number of labels).

For instance on the Wikipedia1K dataset, training a single binary classifier takes about 12.35s on average (for a fixed hyperparameter), and performing prediction on the whole test set takes about 3.18s (for a single classifier, on average). This means that training and testing for BR respectively take about 206 min (1000 x 12.35s) and 53 min, while training and prediction of the individual binary classifiers for RBF with a code size of 500 takes half the time (and thus 1 tenth of the time for a code size of 100). In contrast to the training time of single classifiers given a fixed hyperparameter, finding the optimal configuration for RBFs in terms (#hubs, cluster size) for every code size takes 8 min in total.

- label set distributions/skewed distributions:
The marginal distribution was removed due to lack of space, we will add some numbers in the dataset description. On the Industries dataset, the max label cardinality is 30, and about 20% of the data have a label set of size greater than or equal to 2. On Wikipedia, the max label cardinality is 5, and about 10% of the data have a label set of size greater than or equal to 2.
The distribution of labels is skewed indeed. We believe, however, that this kind of distributions appears in most (if not all) real-life datasets with a large number of classes, as discussed in Dekel et al. Our method is designed to perform well when reasonable label clusters can be found (which we do not believe to be a major limitation in real-life datasets), and, implicitly, when the classes that appear rarely can be predicted better than what a trivial classifier (which never predicts any label) can do (otherwise, Dekel et al.'s baseline is essentially unbeatable). The Dekel et al. baseline is here to show the datasets satisfy the second, implicit requirement, and that the methods actually predict something non-trivial.

- choice of the Hamming loss: We first would like to point out that the Hamming loss is the usual evaluation metric in the context of the paper (multilabel, large number of classes). We believe that the (small) scale of the Hamming loss is not important and that it does not invalidate it. The Hamming loss still gives the natural quantification of "how much do we do compared to a trivial solution". In the multilabel case that we deal with, the trivial solution is to predict no label for every test example. Roughly speaking, the different methods perform between 18% and 36% better than this trivial solution, which is a very substantial improvement. Standard deviations (which, indeed, should and will be added to the result tables) basically follow the small scales of the Hamming loss, and, on our 10-run experiments, the order of magnitude of the standard deviation is 10^-4 times smaller than the loss, for all methods, including the variance in the label clustering, the hash functions, etc. So on the Industries dataset, for a code of size 200, the difference (in hamming loss) between 0.217% (+/- 0.13e-04%) for RBF and 0.205% (+/- 0.29e-04%) for SBF is significant. Note that such low variances are in concordance with previous results (see e.g. [15]).

- Assigned_reviewer7:

(We suppose the reviewer means [12] instead of [9]) The choice of the decoding algorithm for compressed censing (CS) ([12]) was somewhat an issue because there is no clear winner in [12] and there was not enough space in the paper to add two or three more baselines in the tables. We chose Orthogonal Matching Pursuit (OMP) because it was the decoding algorithm used by [15] in their comparisons, and also because OMP seemed to perform well on the precision@k metric in [12] (while correlation decoding (CD) is the worst in terms of precision@k on three cases out of four in Figure 4 of [12]). On the other hand, CD works well in [12] as pointed out by the reviewer, but the good performances are only for the root mean squared error, which is not really reliable as a measure of classification accuracy.

We performed experiments with LASSO and CD, and the results are indeed better with OMP. While LASSO achieves similar performances than OMP for the Hamming Loss, CD is significantly worse in terms of Hamming loss, and both methods have significantly lower (micro/macro-F1). For instance, on Wikipedia1K with a code size of 500, OMP achieves a Hamming loss of 0.0875% and a micro-F1 of 58.5%, while LASSO has a Hamming loss of 0.0898% (micro-F1: 46.8%) and CD has a Hamming loss of 0.100% (micro-F1: 52.3%). We will add such concrete numbers in the text.